# FedTMOS: Efficient One-Shot Federated Learning with Tsetlin Machine

**Shannon How Shi Qi, Jagmohan Chauhan, Geoff V Merrett, Jonathan Hare**
University of Southampton, UK
{s.how,j.chauhan}@soton.ac.uk,{gvm,jsh}@ecs.soton.ac.uk

## Abstract

One-Shot Federated Learning (OFL) is a promising approach that reduce communication to a single round, minimizing latency and resource consumption. However, existing OFL methods often rely on Knowledge Distillation, which introduce server-side training, increasing latency. While neuron matching and model fusion techniques bypass server-side training, they struggle with alignment when heterogeneous data is present. To address these challenges, we proposed One-Shot Federated Learning with Tsetlin Machine (FedTMOS), a novel data-free OFL framework built upon the low-complexity and class-adaptive properties of the Tsetlin Machine. FedTMOS first clusters then reassigns class-specific weights to form models using an inter-class maximization approach, efficiently generating balanced server models without requiring additional training. Our extensive experiments demonstrate that FedTMOS significantly outperforms its ensemble counterpart by an average of $6.16\%$, and the leading state-of-the-art OFL baselines by $7.22\%$ across various OFL settings. Moreover, FedTMOS achieves at least a $2.3\times$ reduction in upload communication costs and a $75\times$ reduction in server latency compared to methods requiring server-side training. These results establish FedTMOS as a highly efficient and practical solution for OFL scenarios.

## 1 Introduction

Federated Learning (FL) enables collaborative training across decentralized data sources while maintaining privacy. Unlike traditional machine learning approaches, where data is transferred to a central server, FL keeps the training process on local devices, with only model parameters being uploaded to a central server for aggregation, ensuring that sensitive information is not shared (McMahan et al., 2016). However, the standard FL training process can lead to significant communication costs due to the multiple communication rounds between the server and devices to achieve convergence. This iterative process is time consuming and resource-intensive, especially for edge devices with limited computational power and bandwidth (Imteaj et al., 2022). The risk to security and the potential of failures in communication links also leads to further challenges (Mothukuri et al., 2021; Li et al., 2020).

To address these limitations, One-Shot Federated Learning (OFL) has emerged as a promising alternative. OFL restricts communication to a single round, thus minimizing communication errors and reducing the risk of interference caused by iterative updates (Guha et al., 2019). This approach is particularly well-suited for scenarios where continuous communication is impractical, such as in model marketplaces, where models are sold after reaching convergence (Li et al., 2020). OFL is ideal for edge devices with limited resources, as it eliminates the need for multiple rounds of transmitting large volumes of model parameters between clients and the server, as well as the need for repetitive on-device training (Khan et al., 2021).

Current OFL methods mainly rely on Knowledge Distillation (KD) and ensemble learning, which aggregate local models into an ensemble before distilling it into a global model. A key challenge with these methods is their dependence on public datasets, which may be inaccessible or unsuitable for certain tasks (Li et al., 2021; Guha et al., 2019). Data-free methods using generative models have been explored (Zhu et al., 2021; Zhang et al., 2022a; Dai et al., 2024; Zhou et al., 2020), but they often suffer from performance limitations and introduce significant computational overhead.

Additionally, neuron matching and model fusion techniques, which eliminate the need for server-side training, have been proposed, but they too struggle with performance when models are trained on heterogeneous data distributions, as aligning the models can be challenging (Wang et al., 2020; Singh & Jaggi, 2020; Jhunjhunwala et al.). On the client side, most existing methods heavily rely on Deep Neural Networks (DNNs), which are resource-intensive and often impractical for clients with limited computational capabilities, such as edge devices.

Firstly, to address the computational limitations of training DNNs on the edge, we employ the Tsetlin Machine (TM), which features a low-complexity architecture based on finite-state automata and game-theoretic principles (Granmo, 2021). This makes TMs a highly efficient alternative to traditional DNNs, as they have demonstrated the capability to significantly reduce communication costs per round by at least $1.4\times$ and improve storage efficiency by at least $6.6\times$ compared to Binary Neural Networks (BNNs), all without sacrificing performance (How et al., 2023).

Furthermore, to enhance the efficiency of the OFL process, we eliminate the need for server-side training for KD and the reliance on an auxiliary dataset. We proposed a novel data-free solution that leverages the class-adaptive nature of TMs by generating server models that maximize inter-class model separation, all without requiring server-side training. Our experiments on various datasets demonstrate that our proposed method outperforms state-of-the-art baselines by an average of $7.22\%$ across all settings. Additionally, it significantly reduces upload communication costs by at least $2.3\times$. By incorporating TMs into OFL, we aim to provide a computationally efficient, scalable, and practical solution for FL scenarios where computational resources are limited.

Our contributions are as follows:

- We introduce FedTMOS, a novel OFL approach that capitalizes on the low-complexity design of the TM. By leveraging TMs, FedTMOS operates as a data-free method that achieves a reduction in upload costs by at least $2.3\times$, while delivering an average accuracy improvement of $7.22\%$ relative to the top-performing baselines and $12.79\%$ improvement over state-of-the-art data-free OFL methods.

- FedTMOS employs a unique inter-class weight separation technique that effectively create server models that enhance class distinction. This approach consistently outperforms its ensemble counterpart by an average of $6.16\%$ across the evaluated datasets, while simultaneously reducing both model size and performance instability by $1.63\%$.

- Our approach significantly reduces server-side computation, achieving at least a $75\times$ reduction in server latency during model aggregation compared to existing methods that rely on server-side training. This efficiency makes FedTMOS highly practical for OFL or situations demanding quick model deployment.

## 2 RELATED WORK

### 2.1 ONE SHOT FL

OFL improves efficiency in FL by limiting communication to a single round. This is typically achieved by building ensemble models and applying KD using public datasets (Guha et al., 2019; Zhou et al., 2020). However, its effectiveness depends on the quality of both the dataset and model ensemble. To enhance performance, Li et al. (2021) applied hierarchical KD, and Diao et al. (2023) introduced open-set voting to generate 'unknown' samples without predefined classes. These methods rely on public data, limiting their use in data-scarce settings. In response, data-free OFL techniques have emerged, including DENSE, which utilizes a generator trained with an ensemble of client models (Zhang et al., 2022a). Dai et al. (2024) further improves the performance of the global model by optimizing through KD derived from both the synthetic data and the ensemble model. Other approaches include local clustering techniques, such as those by Dennis et al. (2021), which involve uploading cluster means instead of full models. While this reduces communication overhead, it struggles with increased data complexity. Heinbaugh et al. (2023) introduced Conditional Variational Autoencoders (CVAEs) to learn conditional data distributions from clients, with the aggregated decoders forming an ensemble model. Several techniques aim to improve server efficiency by avoiding server-side training altogether. Neuron matching aligns model weights (Singh & Jaggi, 2020; Wang et al., 2020), while model fusion combines multiple models into a single robust one

(Jin et al., 2023). These methods uses optimal transport (Singh & Jaggi, 2020), regularization (Jin et al., 2023) and Fisher Information (Jhunjhunwala et al.), facilitate model aggregation without the overhead of KD, enhancing scalability in OFL. However, these methods often struggle with heterogeneous data, limiting their effectiveness.

## 2.2 TMs in FL

FedTM is the first FL framework to utilize TM. Unlike traditional FL frameworks that rely on DNNs, where weight aggregation typically involves a simple weighted average of integer weights, FedTM uses a distinct two-step aggregation process (How et al., 2023). This process is enabled by the unique architecture of the TM, as depicted in Figure 2. While FedTM offers significant reductions in both memory utilization and communication costs, its aggregation process still requires multiple communication rounds to achieve convergence. To our best knowledge, TMs have not been explored in the context of OFL.

# 3 Preliminaries

## 3.1 The Tsetlin Machine

The Tsetlin Machine (TM) is a machine learning method grounded in propositional logic and bit-based representation, leveraging Tsetlin Automata (TA) and game theory principles to derive logical propositions for classification (Granmo, 2021).

### 3.1.1 The Tsetlin Automaton

The Tsetlin Automaton (TA) is an efficient and simple learning mechanism. With $2A$ states, and two actions, Include and Exclude, the automaton adjusts its state based on feedback by either incrementing or decrementing the states based on received feedback (Penalty or Reward). From Figure 1, when a reward is given, the TA advances further along the path of the current action. If it receives a penalty, it moves closer to the center, which may lead to a change in action.

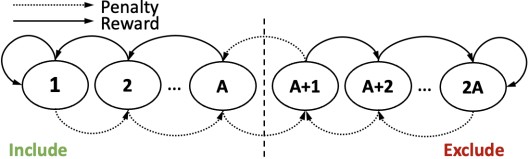

Figure 1: A two-action TA with $2A$ states

### 3.1.2 The Tsetlin Machine Structure and Inference

The first step in training a TM is to convert the input features into a Boolean format. The input feature vector $\mathbf{x} = \{x_1, \ldots, x_o\} \in \{0,1\}^o$ is transformed into a set of literals, $L = \{l_1, \ldots, l_{2o}\}$ $= \{x_1, \ldots, x_o, \neg x_1, \ldots, \neg x_o\}$, which includes both the original features and their negations. Each clause $C_j$ in the TM selects a subset of these literals, denoted as $L_j \subseteq L$. Here, $j$ indexes the clauses, and $L_j$ refers to the set of literals chosen by clause $C_j$ to form its conjunctive expression.

The TM organizes its clauses into two groups: positive and negative clauses. For the given input $\mathbf{x} = \{x_1, \ldots, x_o\}$, and with binary class labels, $y \in \{0,1\}$, the TM computes a unit step function, $u$, to determine the final classification output. If the signed sum $s(\mathbf{x})$ is negative, the TM classifies the output as $\hat{y} = 0$, if not it classifies it as $\hat{y} = 1$:

$$\hat{y} = u(s(\mathbf{x})) = u(\sum_{j=1}^{N/2} C_j^+(\mathbf{x}) - \sum_{j=N/2+1}^{N} C_j^-(\mathbf{x}))$$

In the Multi-Class case, each class has its own set of clauses and the final classification is the class with the highest sums: $\hat{y} = \arg\max_{m=1\ldots M} s_m(\mathbf{x})$. As shown in Figure 2, the TM utilizes bitwise operations to compute its output, resulting in a design that is both intuitive and low in complexity.

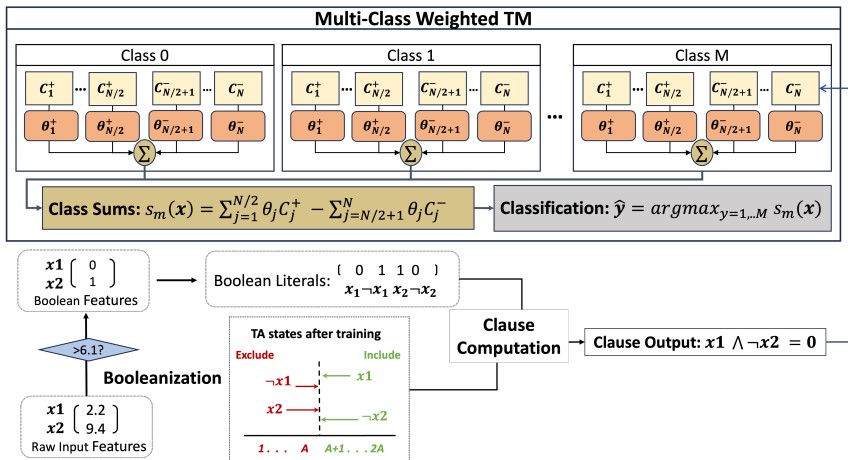

Figure 2: Overview of the Weighted Multi-Class Tsetlin Machine Inference Process: Raw input features are first booleanized to generate boolean literals. Clause outputs are produced through logical operations with these literals and trained TAs. The inference routine sums up the $N$ clause outputs, weighted by their respective weights, to generate class sums for $M$ classes, with the output determined to be the class with the highest sum (Maheshwari et al., 2023).

### 3.1.3 THE TSETLIN MACHINE LEARNING MECHANISM

The TM operates in an online fashion, learning one training example $(\mathbf{x}, y)$ at a time. For each training example, the TM adjusts the states of its TAs by applying rewards and penalties, which involve incrementing or decrementing their states. Feedback to the TAs comes in two types:

- Type I which helps identify frequent patterns is applied randomly to clauses based on their polarity and the true label. Positive clauses receive Type I feedback when $y = 1$, and the negative clauses receive it when $y = 0$.

- Type II is also given stochastically to clauses to enhance pattern discrimination. Positive clauses receive Type II Feedback when $y = 0$, and negative clauses receive it when $y = 1$.

The feedback is applied randomly to introduce an ensemble effect, guided by a hyperparameter $T$, which represents the target sum. The TM aims to adjust the signed sum such that it reaches $-T$ for inputs with class $y = 0$ and $T$ for class $y = 1$. To facilitate this, $s(\mathbf{x})$ is constrained within $[-T, T]$ with: $c(\mathbf{x}) = clamp(s(\mathbf{x}), -T, T)$ and the probability that each clause receives feedback is proportional to the difference between the clamped sum and T:

$$p_y(\mathbf{x}) = \begin{cases} \frac{T+c(\mathbf{x})}{2T}, & \text{if } y = 0 \\ \frac{T-c(\mathbf{x})}{2T}, & \text{if } y = 1 \end{cases} \tag{1}$$

Randomly selecting clauses helps distribute feedback across a variety of significant sub-patterns instead of concentrating on a few. Feedback decreases as the clamped sum, $c(\mathbf{x})$ approaches the target $\pm T$, ensuring that only some clauses are used to identify each sub-pattern.

### 3.1.4 WEIGHTED TSETLIN MACHINE

During learning, similar clauses tend to appear multiple times in the final model. Introducing weights allows each clause to be represented once with an associated weight, rather than repeating it. Hence, the impact of individual clauses can be quantified, resulting in a real-valued quantity.

Initially, all weights are set to 1 and they are updated based on the feedback type. In summary, Type I feedback increases weights for correct patterns, while Type II feedback decreases weights to reduce false positives. This enhances the efficiency of the TM by optimizing weight assignment to clauses, leading to more compact models without compromising accuracy (Phoulady et al., 2020).

The resulting overall sum now, denoted as $s(\mathbf{x})$, becomes:

$$s(\mathbf{x}) = \sum_{j=1}^{N/2} \theta_j^+ C_j^+(\mathbf{x}) - \sum_{j=N/2+1}^{N} \theta_j^- C_j^-(\mathbf{x}) \tag{2}$$

### 3.1.5 CONVOLUTIONAL TM

Convolutional TM (CTM) uses filters with spatial dimensions $W \times W$ and $Z$ binary layers. An image with dimensions $X \times Y$ and $Z$ binary layers is represented in TMs using an input vector $\mathbf{x} = \{x_k \mid k \in \{0,1\}^{X \times Y \times Z}\}$. The input vector represents a specific patch of the image, with the entire image being divided into $B$ such patches. Each clause receives $B$ literal inputs and unlike regular TMs, where a single output is produced per clause, a clause in a CTM outputs $B$ outputs - one for each patch. To aggregate these multiple outputs from clause $j$, $c_j^1, \ldots, c_j^B$ into a a single output, $C_j$, a logical OR operation is applied: $C_j = \bigvee_{b=1}^{B} c_j^b$

Training in the CTM extends the TM's learning process by using both Type I and Type II feedback mechanisms. To update a clause during training, the CTM randomly picks a patch from the set of patches where the clause evaluates to 1, where $\{\mathbf{X}^b \mid c_j^b = 1, 1 \leq b \leq B\}$. The clause is then updated according to the selected patch, allowing the learning process to focus on the most relevant regions that contribute to the clause's outcome (Granmo et al., 2019).

### 3.1.6 TM COMPOSITES

To enable collaboration among multiple independently trained TM models, Granmo (2023) proposed TM composites. Given $r$ TMs, this involve computing the normalized class sums for each TM, $t$ ,by dividing it with the difference between the maximum and minimum class sums in the input set: $\alpha^t = \max_m(s_m^t(\mathbf{x})) - \min_m(s_m^t(\mathbf{x}))$

The final classification result is obtained by selecting the class with the highest value from the sum of all $r$ TMs as computed below:

$$\hat{y} = \arg\max_m \left( \sum_{t=1}^{r} \frac{1}{\alpha^t} s_m^t(\mathbf{x}) \right) \tag{3}$$

TM composites enhance classification accuracy and convergence by reducing over-fitting and increasing robustness among individual models, thereby improving overall performance through effective model combination (Granmo, 2023).

## 4 METHODOLOGY

Given $J$ clients, each having local datasets $D_1, D_2, ..., D_J$. The objective is to aggregate the local TM models, $\mathbf{T} = \{T_1, T_2, ..., T_J\}$, into $\phi$ server models ($\phi < J$) that generalizes well over $\mathbf{D} \equiv \cup_{i \in \mathcal{J}} D_i$ in one communication round.

### 4.1 MOTIVATION

Similar to ensemble methods for OFL, we first introduce a straightforward approach for aggregating local TM models, FedTMOS (ensemble), by applying the principles behind TM Composites. In this approach, the final classification is then determined by Equation 3.

However, this method encounters significant limitations. Firstly, as the number of clients, $J$ increases, the number of local models grows proportionally, leading to issues related to redundancy and overlapping knowledge. This results in diminishing classification performance due to conflicting information among models. Additionally, the increased number of models adds computational overhead, making the OFL process less efficient.

## 4.2 FEDTMOS: ONE-SHOT FEDTM

To address these limitations, we proposed a novel approach based on the class-adaptive nature of TM as illustrated in Figure 2. Clauses, as simple logical expressions, collectively represent complex patterns, with their associated weights quantifying these patterns. Our method aims to reduce the total number of models to a user-defined value $\phi$, where the final classification is determined by:

$$\hat{y} = \arg\max_m \left( \sum_{i=0}^{\phi} \frac{1}{\alpha^i} s_m^i(\mathbf{x}) \right)$$

where $\phi < J$. Our proposed approach consists of two stages: first the weights, represented by $\boldsymbol{\theta} = \{\boldsymbol{\theta}_1, \boldsymbol{\theta}_2, \ldots, \boldsymbol{\theta}_J\}$, where each $\boldsymbol{\theta}_j = \{\theta_j^1, \theta_j^2, \ldots, \theta_j^M\}$ is a $M-$size vector of class-specific weights for client $j$, and $M$ denotes the total number of classes, are scaled based on the average normalized Gini index of all clients ($x$), followed by k-means clustering (Lloyd, 1982). Then, we perform a greedy reassignment of model weights to $\phi$ number of models to minimize overlap.

Our approach is intuitive: by maximizing inter-class separation within models, we enhance the model's ability to distinguish between classes. The Gini index is employed to quantify the distribution of data and ensure balanced client participation. It dynamically adjusts client weights based on data inequality, promoting fairness and engagement across clients (Li et al., 2023). This, in turn, enhances both model performance and convergence.

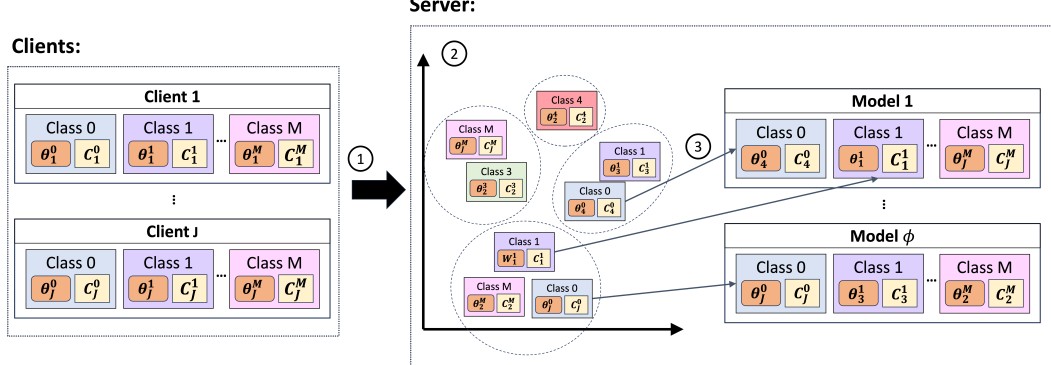

Figure 3: **Overview of FedTMOS:** ① Clients upload their scaled clause weights, state parameters, and normalized Gini Index to the server. ② The server then rescales the weights using the mean normalized Gini Index and performs k-means clustering on the weights. This clustering is essential for grouping similar weights, helping to prevent large disparities in class weights across models during the reassignment process while also reducing complexity. ③ Finally, the number of models, denoted by a user-defined parameter $\phi$, is initialized, and class weights from each cluster are greedily reassigned to maximize inter-class separation within each model.

### 4.2.1 PRE-PROCESSING MODEL WEIGHTS

In the initial step, clients upload their scaled clause weights, which are adjusted based on the proportion of samples per class relative to the client's total sample size. This scaling is expressed as:

$$\theta_j^i = \frac{|D_j^i|}{|D_j|} \theta_j^i$$

where $\theta_j^i$ represents the clause weight for class $i$ in client $j$, $|D_j^i|$ is the number of samples for class $i$ in client $j$, and $|D_j|$ is the total number of samples in client $j$. This ensures that the contributions of different classes are appropriately balanced, preventing over-representation of any particular class.

Alongside these weights, clients also upload their individual normalized Gini index, which quantifies the inequality in their local data distributions (Tangirala, 2020). The Gini Index for client $j$ is

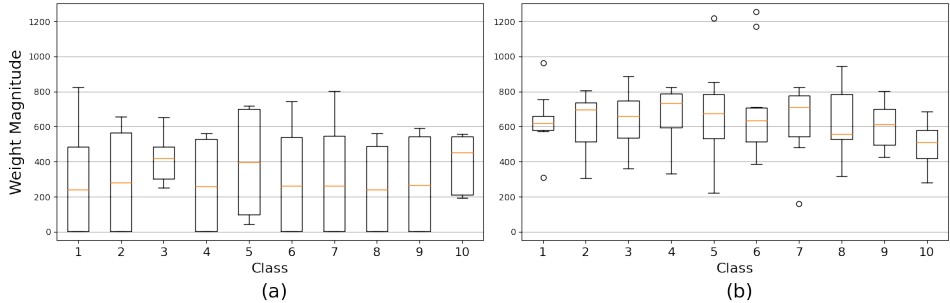

Figure 4: Distribution of clause weights for each class stays within the range of T with similar means for (a) data with low normalized Gini scores compared to (b) data with high normalized Gini scores for a simple TM model with $T = 1000$ on the MNIST dataset.

computed as $g_j = \sum_{i=1}^{M} p_i^2$, where $p_i$ is the proportion of data belonging to class $i$. To account for imbalances in the server model, we scale the clause weights of all clients, $\theta$ using the normalized Gini Index, $x = \frac{1}{J} \sum_{j=1}^{J} g_j$. This scaling adjustment is applied only when $x$ exceeds a predefined threshold, signaling significant inequality. The adjusted weights are updated as:

$$\theta_j^i = x \cdot \theta_j^i$$

This adjustment prevents under-representation of minority classes, which could otherwise be over-shadowed by dominant classes due to imbalanced clause weights as seen in Figure 4. Without correction, these class imbalances, amplified by the signed clamping in TM within $[-T, T]$, can skew the model by decreasing the contribution of the other classes in the server model.

Next, we apply k-means clustering to group these scaled weights. This is crucial as clustering similar weights ensures a smoother and more balanced reassignment process, preventing large disparities in class weights across models. The objective of k-means is to iteratively minimize the within-cluster sum of squares (WCSS), defined as:

$$\underset{\mathbf{V}}{\arg\min} \sum_{i=1}^{k} \sum_{\boldsymbol{\theta} \in V_i} |\boldsymbol{\theta} - \boldsymbol{\mu}_i|^2 \tag{4}$$

where $k$ is the number of clusters, $\boldsymbol{\theta}$ represents a data point (i.e., a vector of scaled weights from any client $j$), $V_i$ is the set of points assigned to cluster $i$, and $\boldsymbol{\mu}_i$ is the centroid (mean) of cluster $i$. Performing k-means on the scaled weights reduces the complexity of the subsequent reassignment process, while ensuring that the class weights within models are not largely skewed.

### 4.2.2 REASSIGNING WEIGHTS

In the initial step, we use k-means clustering to group scaled weights into k clusters, minimizing the WCSS. This approach clusters similar weights, enabling balanced reassignment and reducing the risk of over-fitting or under-representing certain classes.

Building on this clustering foundation, we reassign the weights to $\phi$ models. During this step, we aim to maximize the average squared distances between the cluster centroids that represent different classes within each model. This step is critical because, our goal is to select class weights from the most distinct clusters to ensure that the models have clear class boundaries and better generalization. By doing so, we encourage greater separation between class representations in each model, which helps to improve the model's class differentiation ability.

The objective function for maximizing the inter-cluster distance is as follows:

$$\text{maximize} \quad \frac{1}{\phi} \sum_{m=1}^{\phi} \sum_{i \neq j} \|\mu_m^i - \mu_m^j\|^2 \tag{5}$$

where $\mu_m^i$ and $\mu_m^j$ are the centroid means for classes $i$ and $j$ within model $m$ that correspond to the clusters to which the class weights belong.

This emphasis on inter-class separation ensures that each model maintains distinct class boundaries, thereby enhancing the model's classification performance.

Starting with the clusters containing the fewest classes, we assign class weights to models based on the assignment that yields the greatest increase in average squared distance (Equation 5). If no model satisfies this criterion, we allocate the weight to the model with the fewest distinct class weights. In the event of ties, we select the model with the least overlapping clusters.

To form the final models, we compute the mean of the clause weights for each class. Since clause weights do not affect the TA state updates (Granmo et al., 2019), the aggregation of TA states can occur independently. To prevent the states, represented by bits, from including all literals due to excessive aggregation, we apply the **TopK** procedure from FedTM (How et al., 2023) with $K = 2$. This involves selecting only the two states from the clients with the highest number of samples per class, indicating greater confidence, and combining them using the bitwise OR operator. This approach prevents the problem of overly inclusive and redundant state representations.

Our method maximizes inter-class separation and ensures adequate class coverage, resulting in well-separated models with distinct class representations. This enhances the models' effectiveness in handling data distribution variations and improves class differentiation, yielding scalable, high-performing models for classification tasks. The overall algorithm is summarized in Algorithm 1

---

**Algorithm 1 FedTMOS**

---

**Input:** Clients' scaled local TM model clause weights $\{\boldsymbol{\theta}_1, \boldsymbol{\theta}_2, \ldots, \boldsymbol{\theta}_J\}$ and states $\{\mathbf{C}_1, \mathbf{C}_2, \ldots, \mathbf{C}_J\}, \in \{T_1, T_2, ..., T_J\}$ their normalized Gini scores, $G = \{g_1, g_2, ..., g_J\}$, number of clusters $k$ for k-means clustering and number of final models $\phi$, scaling threshold, $\sigma$
    **Initialize:** all_weights, all_states = [], []
    **Compute** mean normalized Gini scores: $\overline{G} = \frac{1}{J} \sum_{i=1}^{J} G$
    **Set** $x = \begin{cases} \overline{G} & \text{if } \overline{G} > \sigma \\ 1 & \text{if } \overline{G} \leq \sigma \end{cases}$
    **for** each client $j = 1, 2, \ldots, J$ **do**
        **for** each class $m = 1, 2, \ldots, M$ **do**
            **Add** $\theta_j^m \cdot x$ to all_weights and $C_j^m$ to all_states
    cluster_info, cluster_means = **kmeans**(all_weights, $k$)
    reordered_models, reordered_means = **reassign_weights**(cluster_info, cluster_means, $\phi$)
    final_models = **average_models**(reordered_models)
    **return** final_models

---

## 4.3 EXPERIMENTS

### 4.3.1 EXPERIMENTAL DETAILS

**Datasets.** We evaluated our approach on four image datasets widely utilized in FL literature: MNIST (Deng, 2012), F-MNIST (Xiao et al., 2017), SVHN (Netzer et al., 2011) and CIFAR-10 (Krizhevsky, 2009). To simulate heterogeneity, we applied two different methods: sampling class priors from a Dirichlet distribution, $Dir(\alpha)$, as described in Hsu et al. (2019), where $\alpha$ controls the degree of heterogeneity in data splits. We also distributed data such that each client possesses samples from only $\beta$ classes, $S(\beta)$. In our experiments, we simulated non-IID settings by using $\alpha = 0.05, 0.1, 0.3$ and $\beta = 2, 3, 4$.

**Baseline Methods.** We compared our method with several FL algorithms representing different approaches for model aggregation: FedAvg (McMahan et al., 2016), the standard iterative FL algorithm evaluated after one communication round; Fed-Oneshot, which ensembles client classifier predictions (Guha et al., 2019); and Distilled-FedOV, which enhances performance using open-set voting and a public dataset for KD (Diao et al., 2023). DENSE and Co-Boosting, both data-free methods, generate data for KD: DENSE by leveraging similarity, stability, and transferability (Zhang et al., 2022a), and Co-Boosting by creating hard samples and re-weighting client models (Dai et al., 2024). Additionally, we compared with three data-free methods that do not involve server-side training: OT-Fusion, which employs a layer-wise model merging method using optimal transport to align neurons and weights (Singh & Jaggi, 2020); RegMean, which minimizes the $l^2$ distance to individ-

Table 1: Performance of the different algorithms across various data partitions

| Dataset | Partition | FedAvg | Fed-OneShot | DENSE | Co-Boosting | Distilled-FedOV | OT-Fusion | RegMean | FedFisher | FedTMOS (Ensemble) | FedTMOS |
|---|---|---|---|---|---|---|---|---|---|---|---|
| MNIST | $Dir(0.05)$ | 32.67±13.90 | 59.80±27.34 | 68.79±14.19 | 83.42±7.77 | 89.34±0.67 | 52.38±4.74 | 76.64±4.57 | 77.14±8.65 | 87.81±11.31 | **93.80±3.81** |
| | $Dir(0.1)$ | 47.24±18.09 | 74.32±17.18 | 82.05±7.33 | 91.73±5.80 | 93.60±0.62 | 66.20±3.70 | 85.25±1.80 | 79.32±3.05 | 92.44±4.22 | **96.60±1.85** |
| | $Dir(0.3)$ | 78.08±9.05 | 94.81±2.39 | 95.61±1.47 | 96.41±0.83 | 97.67±0.15 | 97.37±0.10 | 94.37±1.42 | 92.03±2.52 | 98.36±0.16 | **98.41±0.10** |
| | $S(2)$ | 24.48±4.72 | 43.45±9.10 | 48.84±12.44 | 62.57±4.09 | 56.26±4.88 | 28.68±7.16 | 65.92±11.88 | 36.92±17.01 | 56.98±0.92 | **92.94±0.51** |
| | $S(3)$ | 35.06±6.82 | 57.30±7.66 | 57.63±8.12 | 79.85±5.55 | 86.22±3.20 | 42.69±3.39 | 69.92±7.57 | 62.66±14.46 | 83.08±0.53 | **95.23±0.47** |
| | $S(4)$ | 51.53±9.00 | 77.02±12.15 | 81.65±7.31 | 93.91±2.48 | 70.92±1.07 | 62.40±11.10 | 87.58±6.06 | 85.12±3.04 | 90.12±0.81 | **96.84±0.50** |
| F-MNIST | $Dir(0.05)$ | 42.14±9.43 | 53.11±2.72 | 51.67±5.31 | 59.93±10.00 | 75.39±0.85 | 42.06±9.06 | 58.02±2.66 | 55.02±6.97 | 68.43±6.08 | **75.45±3.58** |
| | $Dir(0.1)$ | 45.20±14.65 | 60.94±11.25 | 56.97±9.78 | 60.55±6.47 | 77.84±1.21 | 50.66±3.66 | 63.27±5.35 | 60.60±4.91 | 78.14±4.97 | **78.21±3.58** |
| | $Dir(0.3)$ | 71.93±1.54 | 80.24±2.79 | 73.95±2.65 | 78.94±1.69 | 83.24±0.58 | 76.49±4.12 | 75.14±1.26 | 75.57±1.69 | 84.60±0.42 | **84.97±1.39** |
| | $S(2)$ | 11.76±3.05 | 26.89±6.30 | 30.73±8.94 | 40.94±1.95 | 56.97±0.71 | 21.53±3.65 | 35.29±9.21 | 32.25±8.06 | 49.88±0.57 | **59.17±1.56** |
| | $S(3)$ | 27.79±4.45 | 49.70±5.57 | 46.35±7.68 | 59.94±2.11 | 74.85±1.06 | 32.48±5.29 | 60.34±5.18 | 46.07±0.91 | 72.01±0.43 | **74.94±1.78** |
| | $S(4)$ | 36.93±8.22 | 54.48±4.83 | 54.00±4.34 | 57.24±5.38 | 61.20±0.42 | 36.81±2.35 | 68.87±2.78 | 65.21±1.88 | 71.59±0.06 | **78.78±0.77** |
| SVHN | $Dir(0.05)$ | 25.05±15.01 | 39.00±20.47 | 38.09±19.61 | 35.62±12.99 | 63.17±0.95 | 35.97±0.13 | 55.52±3.07 | 56.32±2.29 | 58.71±11.36 | **63.54±2.61** |
| | $Dir(0.1)$ | 35.63±10.05 | 55.06±13.23 | 52.45±11.26 | 53.42±14.28 | 64.94±6.22 | 47.68±1.28 | 55.28±3.36 | 54.35±2.84 | 66.79±1.88 | **69.79±2.33** |
| | $Dir(0.3)$ | 52.99±5.77 | 72.37±6.48 | 65.65±8.95 | 76.09±6.10 | 77.53±2.30 | 77.27±0.21 | 72.43±3.25 | 76.90±1.04 | 75.36±3.31 | **78.86±1.27** |
| | $S(2)$ | 18.51±6.07 | 27.79±5.24 | 26.60±5.78 | 43.29±5.54 | 54.26±8.80 | 17.63±3.61 | 33.25±6.73 | 34.57±6.66 | 43.58±0.72 | **61.63±1.13** |
| | $S(3)$ | 31.31±1.10 | 47.48±5.26 | 43.96±5.47 | 58.89±4.17 | 74.98±0.75 | 29.67±7.82 | 54.66±6.73 | 57.25±3.96 | 72.83±0.23 | **78.09±0.25** |
| | $S(4)$ | 43.24±3.03 | 56.35±2.51 | 51.44±4.34 | 63.88±5.39 | 73.47±0.56 | 35.69±4.41 | 63.89±2.54 | 64.77±6.53 | 72.92±0.35 | **75.24±0.68** |
| CIFAR-10 | $Dir(0.05)$ | 17.02±0.83 | 28.30±5.68 | 25.97±2.52 | 26.17±3.85 | 40.04±5.61 | 30.70±0.80 | 33.58±4.59 | 35.61±1.41 | 37.38±13.21 | **47.69±2.59** |
| | $Dir(0.1)$ | 27.55±5.30 | 37.52±4.53 | 32.46±3.62 | 36.71±7.87 | 46.79±2.66 | 31.46±1.52 | 33.49±0.83 | 39.55±5.36 | 45.85±10.00 | **52.25±1.71** |
| | $Dir(0.3)$ | 36.44±2.45 | 50.83±2.99 | 43.11±0.95 | 43.35±2.08 | 48.24±5.22 | 49.43±2.94 | 44.34±1.42 | 51.18±1.45 | 56.96±0.88 | **57.03±0.74** |
| | $S(2)$ | 15.93±5.23 | 13.95±5.86 | 15.73±3.77 | 21.91±6.51 | 33.90±0.12 | 18.57±0.59 | 24.24±2.57 | 22.23±1.53 | 38.47±0.44 | **39.95±1.33** |
| | $S(3)$ | 25.45±0.75 | 35.83±4.98 | 29.13±3.26 | 33.21±4.00 | 45.64±2.14 | 24.70±2.77 | 31.06±1.36 | 32.36±4.15 | 51.53±0.42 | **52.18±0.73** |
| | $S(4)$ | 24.96±0.63 | 39.04±3.41 | 31.26±1.61 | 36.39±1.22 | 41.17±1.63 | 33.67±0.62 | 38.97±1.33 | 41.46±2.31 | 59.41±1.36 | **59.59±0.30** |

ual model predictions using inner product matrices of layer inputs (Jin et al., 2023); and FedFisher, which approximates Bayesian inference using Fisher information by using local posterior estimates and Fisher matrices from clients (Jhunjhunwala et al.). These baselines were selected to evaluate the performance with and without reliance on external datasets or excessive computational overhead.

**Configurations.** Our experiments default to 10 clients with $Dir(0.1)$, as in Zhang et al. (2022a); Dai et al. (2024), unless stated otherwise. We reported the average test accuracy and standard deviation across 3 different dataset splits for each setting. Following Diao et al. (2023), half of the test dataset served as a public dataset for Distilled FedOV, and we evaluated all algorithms on the same subset, ensuring consistency across evaluations. For CNN-based algorithms, we used a 5-layer CNN architecture with a batch size of 128, as per Dai et al. (2024) and Jhunjhunwala et al.. For FedT-MOS, we fixed $k = 30$ for the $Dir$ partitions to account for class distribution variability among clients and $k = 10$ for the $S$ partition. We implemented different CTM models for each dataset, ensuring the size of $\phi$ server models was equal to or smaller than the distilled CNN server model for fair comparison. All algorithms were trained for 30 local epochs, as outlined in Jhunjhunwala et al.. See Appendix A.1 for more details.

From Table 1, FedTMOS outperforms all baseline methods without the need for synthetic or generated data and server-side training in all settings. On average, FedTMOS performs better than the top performing baseline, Distilled-FedOV, by $13.3\%, 3.67\%, 3.13\%, 8.82\%$ on the MNIST, F-MNIST, SVHN and CIFAR-10 dataset respectively. We note that Distilled FedOV leverages KD at the server using a subset of the test data, closely matching the actual samples. On average, FedTMOS outperforms the best data-free OFL method by $12.79\%$, reinforcing its effectiveness in OFL.

We note that DENSE and Co-Boosting exhibit high variance, particularly with increasingly non-IID data. This is likely because classifiers trained on non-IID data generate sub-optimal knowledge samples, which are crucial for the distillation process. This hinders the student model's learning, reducing performance and increasing variability (Gou et al., 2021).

We also evaluated FedTMOS against the FedTMOS ensemble. On average, FedTMOS demonstrates $6.16\%$ better performance across all settings, while reducing variation by an average of $1.63\%$ across all settings. Furthermore, FedTMOS exhibits lower memory costs as it reduces the ensemble model size to $\phi$ number of server models, making it an efficient solution. Overall, these results highlight FedTMOS as a reliable approach for data-free OFL.

### 4.3.2 EFFICIENCY ANALYSIS

In terms of communication costs, our results demonstrate that the bit-based architecture of the TM significantly reduces communication overhead while producing a model that generalizes well across all settings, evident in the results in Table 2. By ensuring that the server model size of the TM is equal to or smaller than the CNN models, we show that FedTMOS effectively reduces upload costs by at least $2.3\times$ without sacrificing performance.

Table 2: CC: Upload costs for each client/storage and potential download costs of the server model (MB); SL: Server aggregation latency (s).

| Dataset | Metric | DENSE | Co-Boosting | Distilled-FedOV | OT-Fusion | RegMean | FedFisher | FedTMOS |
|---------|--------|-------|-------------|-----------------|-----------|---------|-----------|---------|
| MNIST | CC | 0.25/0.25 | | | 0.18/0.18 | | | **0.04/0.16** |
| | SL | 647±3.42 | 738±5.76 | 303±14.86 | 0.71±0.05 | 2.49±00.34 | 11.06±1.45 | **0.48±0.09** |
| F-MNIST | CC | 0.25/0.25 | | | 0.18/0.18 | | | **0.05/0.14** |
| | SL | 624±2.75 | 835±3.38 | 317±7.53 | 0.78±0.03 | 2.96±0.19 | 11.2±1.06 | **0.52±0.14** |
| SVHN | CC | 1.28/1.28 | | | 3.76/3.76 | | | **0.31/1.23** |
| | SL | 694±3.40 | 1238±4.52 | 962±7.18 | 2.29±1.31 | 8.19±1.62 | 13.23±1.42 | **1.21±0.25** |
| CIFAR-10 | CC | 1.28/1.28 | | | 3.76/3.76 | | | **0.57/1.24** |
| | SL | 931±3.51 | 1247±7.45 | 246±5.95 | **2.04±1.14** | 6.93±0.69 | 9.98±0.93 | 3.24±0.17 |

We evaluated the average latency for model aggregation at the server using a standard compute node equipped with a single GPU core. FedTMOS consistently outperforms other algorithms by achieving lower server aggregation latency. However, for the CIFAR-10 dataset, OT-Fusion performs slightly better, with a latency that is 1.2 seconds faster than FedTMOS. Despite this, FedTMOS still achieves, on average, 20% better performance than OT-Fusion on CIFAR-10, suggesting that FedTMOS is more efficient overall, even if its latency is marginally higher in this specific case.

Overall, these results indicate that FedTMOS has significant potential for efficiency, attributed to its low upload costs, which is essential for edge devices with limited bandwidth. Furthermore, reduced aggregation latency ensures that the OFL process remains efficient, regardless of the computational resources available on the aggregator server. This is particularly important in edge-based FL, where an edge server acts as an aggregator managing nearby edge devices. The absence of server-side training contributes to quicker deployment in time-sensitive scenarios and allows for the extension to iterative FL while maintaining efficiency (Zhang et al., 2022b).

### 4.3.3 SCALABILITY

We evaluated the performance of all methods by varying the number of clients. For FedTMOS, to constrain the server model size, we reduced the number of clauses in the local CTM model, scaling down the local model size by $2.9\times$ for 20 and 50 clients, and $4\times$ for 80 clients. From Table 3, we see that FedTMOS outperforms all other methods, although its performance declines with an increasing number of clients. This suggests that while reducing model size is necessary, it can limit performance. Future work will explore dynamic scaling techniques for the number of server models, $\phi$, and adapting the number of clusters, $k$, to better balance complexity and class coverage. Additionally, advanced sampling methods can be used to manage the increasing number of weight vectors for reassignment, potentially enhancing performance with increasing clients.

Table 3: Performance of the different algorithms with an increasing number of clients on CIFAR-10

| Clients | FedAvg | Fed-OneShot | DENSE | Co-Boosting | Distilled-FedOV | OT-Fusion | RegMean | FedFisher | FedTMOS |
|---------|--------|-------------|-------|-------------|-----------------|-----------|---------|-----------|---------|
| 20 | 26.42±7.99 | 26.64±15.74 | 31.18±6.90 | 34.55±3.89 | 37.20±1.55 | 24.95±3.24 | 18.55±8.92 | 32.46±2.34 | **50.08±3.62** |
| 50 | 23.34±4.04 | 34.96±3.60 | 27.47±3.77 | 34.33±2.08 | 30.43±1.56 | 27.17±1.36 | 31.74±1.94 | 37.52±0.37 | **50.90±0.94** |
| 80 | 23.04±5.11 | 25.04±13.06 | 23.85±11.70 | 32.47±2.36 | 25.65±1.04 | 22.20±1.85 | 20.86±2.25 | 33.66±0.95 | **49.15±1.08** |

## 5 CONCLUSIONS

We introduced FedTMOS, a novel framework for OFL that eliminates the need for server-side training for KD and the use of synthetic or generated datasets. By scaling weights with normalized Gini scores, clustering parameters, and maximizing inter-class model separation, FedTMOS reduces computational complexity, constrains server model size, and minimizes variance in performance. Notably, it surpasses all SOTA baselines by at least an average of 7.22% and the best data-free method by 12.79% across all dataset settings. Furthermore, it achieves a reduction in upload communication costs by at least $2.3\times$, making FedTMOS well-suited for FL with edge devices and providing a strong foundation for further exploration into enhancing the efficiency and performance of OFL.

## 6 ACKNOWLEDGMENTS

This work was supported by the UK Research and Innovation (UKRI) Centre for Doctoral Training in Machine Intelligence for Nano-electronic Devices and Systems [EP/S024298/1] and the Engineering and Physical Sciences Research Council (EPSRC) International Centre for Spatial Computational Learning [EP/S030069/1].

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

## A   APPENDIX

### A.1   ADDITIONAL EXPERIMENTAL DETAILS

#### A.1.1   BASELINES MODEL CONFIGURATION

For the baseline models, we adhered to the same hyperparameters specified in the original papers (Jhunjhunwala et al.; Zhang et al., 2022a; Dai et al., 2024; Diao et al., 2023). However, to ensure a fair comparison, we fixed the number of server epochs to 100 epochs for the baseline algorithms - Distilled FedOV, DENSE and Co-Boosting.

#### A.1.2   FEDTMOS MODEL CONFIGURATION

To align with the booleanized format for training TMs, we applied variouos pre-processing steps for each dataset. For the MNIST dataset, we encoded the data by setting pixel values larger than 75 to 1, and values below or equal to 75 to 0. For the F-MNIST and SVHN dataset, we followed the original implementation by Granmo et al. (2019), to binarize the data using an adaptive Gaussian thresholding procedure with a window size of 11 and a threshold value of 2. For CIFAR-10, as described in Granmo (2023), we processed one copy of the data using the same adaptive Gaussian thresholding procedure and another copy with 8-level color thermometer encoding. Additionally, we employed three TM composite models for CIFAR-10: adaptive thresholding, 3x3 color thermometer, and 4x4 color thermometer.

For FedTMOS, we used the following model configurations of CTM for each dataset:

Table 4: FedTMOS Model Configuration

|  | MNIST | F-MNIST | SVHN | CIFAR-10 | | |
|---|---|---|---|---|---|---|
|  |  |  |  | Adaptive Threshold | 3x3 CT | 4x4 CT |
| Number of Clauses | 100 | 200 | 1100 | 200 | 200 | 200 |
| Feedback Threshold | 1000 | 1000 | 2000 | 400 | 300 | 300 |
| Learning Sensitivity | 5 | 5 | 5 | 5 | 5 | 5 |
| Patch Dimensions | (10,10) | (5,5) | (5,5) | (10,10) | (3,3) | (4,4) |

#### A.1.3   ADDITIONAL SETTINGS

For FedTMOS, as we wanted to constrain our models such that the final server model aligns with smaller or equal sizes to the CNN counterparts, we set various $\phi$ values to constrain our final model. For the MNIST, F-MNIST, SVHN and CIFAR-10 dataset we used : $\phi \leq \{4, 3, 3, (3, 3, 1)\}$ respectively. Additionally, we employed threshold values based on the average normalized Gini Index ($\overline{G}$) to scale the clause weights. When $\overline{G}$ exceeds a threshold, $\sigma$, it indicates significant class imbalance. In response, clause weights are scaled by $x$ to prevent dominant classes from overpowering minority ones, ensuring balanced class representation in the final model and mitigating the effects of non-IID data. The code is available at: https://github.com/shannonhsq/FedTMOS.

$$x = \begin{cases} \overline{G} & \text{if } \overline{G} > \sigma \\ 1 & \text{if } \overline{G} \leq \sigma \end{cases}$$

We used threshold = $\{0.5, 0.5, 0.3, 0.6\}$ respectively for the MNIST, F-MNIST, SVHN and CIFAR-10 dataset. Note that we used $0.3$ for SVHN due to the unbalanced nature of the data. In the future, we will explore a data-driven approach to determine the threshold and $k$ values.

### A.2   ADDITIONAL RESULTS

#### A.2.1   PERFORMANCE WITH 200 LOCAL EPOCHS

We evaluated the baseline algorithms with 200 local epochs, as recommended in (Dai et al., 2024), to ensure that local models reached convergence before server-side training. However, we excluded

Table 5: Performance of the different algorithms across various data partitions

| Dataset | Partition | FedAvg | Fed-OneShot | DENSE | Co-Boosting | Distilled-FedOV | FedTMOS (Ensemble) | FedTMOS |
|---|---|---|---|---|---|---|---|---|
| MNIST | $Dir(0.05)$ | 38.65±17.31 | 41.78±11.34 | 78.36±8.69 | 88.95±3.89 | 89.25±3.92 | 83.78±11.24 | **93.50±3.72** |
| | $Dir(0.1)$ | 48.63±12.77 | 56.12±13.34 | 85.15±5.94 | 94.88±0.49 | 93.83±3.02 | 93.86±2.09 | **97.28±1.31** |
| | $Dir(0.3)$ | 80.01±10.18 | 88.25±5.38 | 96.19±1.07 | 96.09±0.62 | 97.56±0.32 | 98.30±0.39 | **98.50±0.12** |
| | $S(2)$ | 21.55±7.39 | 24.23±3.21 | 53.72±15.88 | 72.22±9.56 | 69.16±7.05 | 48.88±9.01 | **83.07±3.51** |
| | $S(3)$ | 35.24±8.86 | 40.45±8.99 | 70.73±10.65 | 81.01±8.04 | 80.23±4.75 | 74.18±4.20 | **88.75±3.46** |
| | $S(4)$ | 40.28±9.74 | 48.66±9.70 | 72.27±5.06 | 87.16±5.35 | 84.08±8.18 | 88.68±0.61 | **97.16±0.48** |
| F-MNIST | $Dir(0.05)$ | 39.89±6.48 | 41.97±5.78 | 62.52±5.28 | 64.58±2.48 | 72.00±3.16 | 70.33±6.39 | **72.73±3.23** |
| | $Dir(0.1)$ | 50.90±8.27 | 50.56±6.66 | 70.07±2.92 | 66.94±3.85 | 78.15±3.15 | 75.38±3.76 | **78.39±3.75** |
| | $Dir(0.3)$ | 71.26±1.03 | 73.14±2.55 | 80.75±0.59 | 80.14±1.49 | 83.31±0.61 | 83.87±0.33 | **84.30±0.69** |
| | $S(2)$ | 24.74±7.32 | 22.78±3.69 | 40.72±6.54 | 48.16±3.24 | 54.72±4.06 | 39.80±7.92 | **60.08±7.67** |
| | $S(3)$ | 44.28±6.35 | 34.62±4.30 | 57.05±4.95 | 52.44±3.81 | 67.81±5.57 | 63.60±1.90 | **72.82±1.61** |
| | $S(4)$ | 34.66±2.01 | 34.73±6.02 | 63.14±7.08 | 67.85±2.53 | 68.68±7.02 | 76.62±6.34 | **80.76±6.19** |
| SVHN | $Dir(0.05)$ | 28.04±18.44 | 24.61±2.88 | 46.14±15.66 | 50.81±13.83 | **70.45±3.00** | 41.45±18.69 | 66.77±1.41 |
| | $Dir(0.1)$ | 40.34±12.91 | 44.04±13.20 | 58.54±8.01 | 62.33±10.32 | **74.31±3.26** | 67.28±1.27 | 72.90±3.51 |
| | $Dir(0.3)$ | 58.00±0.82 | 76.29±4.15 | 67.80±3.94 | 76.52±4.79 | **81.35±1.01** | 76.41±1.10 | 80.73±0.57 |
| | $S(2)$ | 18.95±1.78 | 24.16±6.06 | 38.42±10.93 | 48.42±4.93 | 62.25±2.88 | 42.84±7.81 | **62.32±2.78** |
| | $S(3)$ | 37.21±5.04 | 38.03±7.75 | 45.17±11.12 | 62.95±1.88 | 72.31±4.57 | 66.43±6.48 | **75.62±3.02** |
| | $S(4)$ | 36.32±10.06 | 41.47±5.64 | 47.66±6.32 | 57.57±7.69 | 76.76±3.04 | 74.34±3.84 | **80.64±0.82** |
| CIFAR-10 | $Dir(0.05)$ | 20.96±1.57 | 26.25±6.18 | 37.80±5.20 | 39.46±5.07 | 44.19±1.96 | 38.02±6.86 | **47.12±4.95** |
| | $Dir(0.1)$ | 27.14±8.51 | 36.05±4.60 | 48.57±3.74 | 49.15±10.56 | 48.89±4.23 | 46.88±4.44 | **52.06±2.77** |
| | $Dir(0.3)$ | 43.99±6.10 | 42.01±3.94 | 60.12±3.36 | **60.72±3.83** | 56.54±1.88 | 58.43±0.08 | 56.69±2.07 |
| | $S(2)$ | 14.71±6.36 | 18.47±2.02 | 29.26±5.25 | 32.94±5.56 | 31.57±6.20 | 32.16±5.06 | **36.01±3.18** |
| | $S(3)$ | 23.01±6.68 | 24.68±4.66 | 40.52±3.50 | 42.15±5.84 | 43.06±1.94 | 50.66±2.82 | **53.21±2.09** |
| | $S(4)$ | 34.66±3.51 | 34.93±5.01 | 41.54±5.12 | 46.91±7.58 | 47.83±2.24 | 57.78±1.57 | **57.85±1.13** |

OT-Fusion, RegMean, and FedFisher from this setup, as their hyperparameters are specifically optimized for 30 epochs according to Jhunjhunwala et al., and re-tuning for 200 epochs could inadvertently introduce inconsistencies.

From Table 5, FedTMOS outperforms the other baseline methods without the need for synthetic or generated data in most settings. On average, FedTMOS performs better than the top performing baseline, Distilled FedOV by $7.36\%, 4.07\%, 0.26\%, 5.14\%$ on the MNIST, F-MNIST, SVHN and CIFAR-10 dataset respectively. We note that Distilled FedOV leverages KD at the server using a subset of the test data, closely matching the actual samples. Furthermore, when compared to the best-performing data-free method for each setting, FedTMOS surpasses them in all except for the least non-IID setting ($Dir(0.3)$) for CIFAR-10. On average, FedTMOS outperforms the best data-free method for each setting by $8.46\%$, reinforcing its effectiveness in data-free OFL.

In addition, Table 1 highlights that even with just 30 epochs, FedTMOS maintains its ability to outperform all baselines. This result emphasizes the effectiveness of FedTMOS, showcasing its strong performance even without requiring full convergence.

We note that DENSE and Co-Boosting exhibit high variance, particularly with increasingly non-IID data. This is likely because classifiers trained on non-IID data generate sub-optimal knowledge samples, which are crucial for the distillation process. This hinders the student model's learning, reducing performance and increasing variability (Gou et al., 2021).

### A.2.2 ADDITIONAL EFFICIENCY ANALYSIS

Table 6: Training Latency (s)

| Dataset | DENSE | Co-Boosting | Distilled-FedOV | OT-Fusion | RegMean | FedFisher | FedTMOS |
|---|---|---|---|---|---|---|---|
| MNIST | | 61.29±0.14 | | | 30.16±2.33 | | 22.21±0.79 |
| F-MNIST | | 61.96±0.41 | | | 48.84±4.15 | | 34.20±3.49 |
| SVHN | | 271.46±4.16 | | | 25.86±1.94 | | 320.54±9.42 |
| CIFAR-10 | | 235.10±6.86 | | | 22.54±2.35 | | 219.58±6.85 |

In terms of communication costs, our results demonstrate that the bit-based architecture of the TM significantly reduces communication overhead while producing a model that generalizes well across

all settings, evident in the results in Table 2. By ensuring that the server model size of the TM is equal to or smaller than the CNN models, we show that FedTMOS effectively reduces upload costs by at least $2.3\times$ without sacrificing performance. Furthermore, FedTMOS reduces storage costs by an average of $3.5\times$ compared to FedTMOS (ensemble), while providing an average of $6.16\%$ improved performance and $1.63\%$ decrease in variability across all settings.

We computed the average training latency of the models on each client with a compute node with 32 CPU cores. From Table 6, even though the training latency of FedTMOS is higher compared to OT-Fusion, RegMean, and FedFisher, the results in Table 1 demonstrate that FedTMOS outperforms them by at least an average of $14.43\%$ across all settings. This substantial improvement in performance highlights the value of FedTMOS in achieving higher accuracy with comparable server efficiency. Moreover, despite the slight trade-off in training efficiency, FedTMOS remains highly suitable for practical deployment due to its low upload costs as shown in Table 2, which are crucial for edge devices with limited bandwidth (Zhang et al., 2022b), and the computational simplicity of TMs, which avoid backpropagation and make them ideal for low-power training on edge devices (Rahman et al., 2022; Tang et al., 2024; Lei et al.). This balance between performance and efficiency highlights FedTMOS as a promising solution for OFL and facilitates its extension to iterative FL while maintaining efficiency (Zhang et al., 2022b).

Furthermore, we observed that the training latency of OT-Fusion, RegMean, and FedFisher is lower than that of DENSE, Co-Boosting, and Distilled-FedOV. This difference arises because the former methods utilize CNNs with three layers (Dai et al., 2024), whereas the latter methods employ CNNs with only two layers (Jhunjhunwala et al.).

### A.2.3 Performance Evaluation in a Centralized Setting

Table 7: Performance of CTM and CNN in the centralized setting, where CNN1 is the CNN in FedAvg, Fed-OneshotDENSE, Co-Boosting, Distilled-FedOV and CNN2 is the CNN in OT-Fusion, RegMean, FedFisher

|  | CTM | CNN1 | CNN2 |
|---|---|---|---|
| MNIST | 98.23±0.12 | 99.15±0.01 | **99.51±0.09** |
| F-MNIST | 87.30±0.56 | **90.04±0.33** | 89.66±0.29 |
| SVHN | 80.07±0.20 | 89.51±0.37 | **90.19±0.59** |
| CIFAR10 | 54.93±0.45 | **82.17±0.43** | 79.42±0.55 |

In a centralized setting, CTM does not outperform CNNs in terms of accuracy as shown in Table. 7. This is expected, as CNNs are highly optimized for tasks like image classification and generally achieve higher performance in such settings (Granmo, 2023).

Note that the performance of CTM in our experiments is lower than that reported in Granmo (2023). This discrepancy arises because, to ensure model size and efficiency constraints, we used only 200 clauses per client compared to 2000 clauses in the original paper. Additionally, we used 3 models per client, whereas the paper utilized 4 models.

However, this highlights that the robustness of our proposed approach, FedTMOS, enables CTM to perform competitively compared to the CNN-based OFL baselines, even though CTM may not match the accuracy of CNNs in centralized settings. The performance improvements observed in FL settings stem from the strength of our proposed OFL methodology, FedTMOS, which is both robust and scalable in FL.

### A.2.4 Extension to Complex Datasets

We compared FedTMOS with ResNet-18 on CIFAR-100 Krizhevsky (2009) and with a pre-trained ResNet-18 model on Tiny-ImageNet Le & Yang (2015). While TMs have yet to be directly explored for the use of pre-trained weights, the potential for applying them to tasks that share classes with the pre-training dataset is promising, as demonstrated by the comparable performance on the Tiny-ImageNet dataset between a pre-trained ResNet-18 and TM. This will be a key focus of our future work, where we aim to investigate how incorporating pre-trained weights can further enhance TM

adaptability and performance. Regardless, the performance of our current model, even without large-scale pre-training, highlights the potential of TMs, suggesting that there is still room for significant improvement in their capabilities.

Table 8: Performance of the evaluated baselines on complex datasets

|  |  | FedAvg | DENSE | Co-Boosting | FedTMOS |
|---|---|---|---|---|---|
| CIFAR-100 | $Dir(0.05)$ | 6.93±1.07 | 20.12±2.40 | 20.16±2.75 | **27.16±0.57** |
|  | $Dir(0.1)$ | 10.52±0.44 | 25.22±2.02 | 25.38±1.53 | **28.19±0.86** |
|  | $Dir(0.3)$ | 13.32±0.76 | **30.97±0.89** | 30.26±0.51 | 30.51±0.98 |
| Tiny-ImageNet | $Dir(0.05)$ | 7.44±0.12 | 8.52±0.32 | 8.38±0.14 | **11.64±0.65** |
|  | $Dir(0.1)$ | 9.41±1.12 | 10.61±0.57 | 10.49±0.19 | **11.81±0.41** |
|  | $Dir(0.3)$ | 12.29±0.39 | 13.85±0.75 | **14.35±0.35** | 13.03±0.66 |

Table 9: Upload costs for each client/storage and potential download costs of the server model (MB)

| Dataset | FedAvg | DENSE | Co-Boosting | FedTMOS |
|---|---|---|---|---|
| CIFAR-100 | 45.12/45.12 | | | **14.2/43.4** |
| Tiny-ImageNet | 45.12/45.12 | | | **11.36/40.8** |

## A.3 FURTHER DISCUSSION ON THE LIMITATIONS AND FUTURE WORK

Firstly, we acknowledge that TMs in general, are not as robust as DNNs for complex datasets, particularly those involving multi-channel images. This limitation stems from the booleanization process and bit-based representation of input data, which restrict TMs' performance. Efforts to address this, such as creating composite TMs trained on extracted features like 3x3 thermometer encoding and Histograms of Gradients, have shown promise in improving their capability (Granmo, 2023).

However, DNNs face significant challenges in FL scenarios with high data heterogeneity. When aggregated at the server, the parameter spaces of clients with heterogeneous data often fail to provide an accurate estimation of the global parameter space. Unlike DNNs, TMs leverage class-specific weights, enabling more effective contributions from individual clients in heterogeneous settings. While class-wise weights could theoretically be implemented in DNNs, TMs are inherently more suited for this purpose due to their simpler structure and efficient aggregation process.

For example, in the MNIST dataset, TMs require just 100 weights per class, compared to a simple CNN, which might involve 61,706 weights (Dai et al., 2024), translating to at least 6,179 weights per class. This simplicity reduces computational overhead. Furthermore, while small DNNs trained for individual classes often risk over-fitting due to limited generalization capacity, TMs are simultaneously trained across all classes with separable class weights. This design helps TMs generalize better across classes, reducing over-fitting and making them more adept at handling heterogeneous data in FL settings. While DNNs might struggle in similar scenarios due to their complexity, exploring class-specific weights in DNNs for FL settings remains an interesting avenue for future research.

Next, TMs remain an emerging field, and the use of pre-trained models has yet to be fully explored. Unlike DNNs, which often benefit from pre-training on large datasets like ImageNet (Chen et al., 2023; Nguyen et al., 2023), TMs have not been adapted to incorporate pre-trained weights. For instance, a TM pre-trained on a large dataset could potentially reuse class-specific weights for tasks involving those same classes. This unexplored area presents an exciting direction for future work. Pre-training a TM on a dataset like ImageNet and fine-tuning it on smaller datasets, such as Tiny ImageNet or CIFAR-100, could significantly enhance their applicability and effectiveness in these specific tasks.

Lastly, in our current experiments, we also observed that the number of resulting weight vectors from local clients can influence recombination performance, especially when mapping to a fixed number of server models. As discussed in the main text, dynamic scaling of the number of server

models, $\phi$, and the number of clusters, $k$, remains a key direction for future work. In particular, we aim to explore how $\phi$ can be dynamically adapted based on the number of available weight vectors, to better balance performance and model complexity.

In summary, while TMs are still an emerging field of research, they offer distinct advantages, particularly in FL scenarios. Their class-specific weight structure and computational simplicity make them well-suited for low-power, on-device training (Tang et al., 2024; Lei et al.; Rahman et al., 2022). However, we recognize their current limitations, including challenges in handling complex datasets and the lack of pre-trained models, which can affect training efficiency.

## A.4 Full Algorithm

---

**Algorithm 2 reassign_weights(cluster_info, cluster_means, $\phi$)**

---

**Initialize:** reordered_models = rm for rm in range($\phi$), reordered_means = {}, used_clusters = [] , track_clusters = {[] for rm in range($\phi$)}
**Sort** cluster_info based on number of classes in each cluster
**while** len(used_clusters) < num_clusters **do**
    **for** cidx in cluster_info **do**
        **if** cidx in used_clusters **then**
            **continue**
        **for** class m in cidx **do**
            best_model = **find_best**(cidx,m,reordered_models, reordered_means)
            **if** best_model == False **then**
                best_model = **find_least_cc**(reordered_models,cidx)
            **Add** $\theta_j^m$ to best_model
            **Add** $C_j^m$ to best_model
            **Update** the mean for class $m$ in reordered_means[best_model][m]
        Add cidx to used_clusters
        Add cidx to track_clusters[best_model]
**return** reordered_models

---

**calculate_avg_link(reordered_means)**:
    ss_2_dist = 0
    num_pairs = 0
    **for** mi in range(len(reordered_means)) **do**
        **for** mj in range(mi+1,len(reordered_means)) **do**
            ss_2_dist $+=$ $\|$reordered_means[mi]-reordered_means[mj]$\|^2$
            num_pairs += 1
    **return** ss_2_dist/num_pairs

**find_best(cidx,m,reordered_models,reordered_means)**:
    best_model = False, max_dist = 1
    **for** model in reordered_models **do**
        **if** model has class m **then**
            **continue**
        temp_model_means = copy(reordered_means[model])
        temp_model_means[model][m].update(reordered_means[cidx])
        distance = **calculate_avg_link**(temp_model_means)
        **if** distance> max_dist **then**
            max_dist=distance
            best_model = model
    **return** best_model

**find_least_cc(reordered_models, cidx)**
    best_model = False, min_class = -1
    **for** rm **in** reordered_models **do**
        distinct_classes = len(set(classes in rm))
        **if** distinct_classes < min_class **then**
            min_class = distinct_classes
            best_model = rm
        **else if** distinct_classes == min_class **then**
            cluster_class_count = **sum**(1 **for** c **in** track_clusters[rm])
            **if** cluster_class_count < min_class **then**
                best_model = rm
    **return** best_model

---

---

**Algorithm 3 average_models(final_models)**

---

**for** each model fm in final_models **do**
    **for** each class m in fm **do**
        sorted_indices $\leftarrow$ argsort($\theta_j^m$ for all $j \in fm$, in descending order)
        $C_{fm}^m \leftarrow \bigvee_{j \in \text{sorted\_indices}[0:2]} C_j^m$
        $\theta_{fm}^m \leftarrow \text{mean}(\theta_j^m), \forall j \in fm$
**return** final_models

---

