# OpenReview forum: "FedTMOS: Efficient One-Shot Federated Learning with Tsetlin Machine"
_ICLR.cc/2025/Conference — ICLR 2025 Poster_

### Official Review · Reviewer_Voa4 · 2024-10-26

**Soundness:** 2
**Presentation:** 1
**Contribution:** 2
**Rating:** 6
**Confidence:** 3

**Summary:**

This paper propose FedTMOS for efficient one-shot federated learning (FL). FedTMOS employs Tsetlin Machine instead of DNNs to reduce upload costs and presents a novel data-free solution to generate server model. Experimental results show that FedTMOS outperforms existing one-shot FL methods.

**Strengths:**

- Employing Tsetlin Machine in one-shot federated learning is interesting.
- The proposed FedTMOS significantly reduce the communication costs.

**Weaknesses:**

- It is unclear whether the performance improvement in Table 1 comes from the performance gap between the CNNs and CTM. It is suggestted to report the performance of CNNs and CTM in a centralized(non-federated learning) setting.
- My main concern with this work is its applicability, as it is limited to a specific machine learning model. In my view, machine learning models and tasks should primarily serve as a testbed for evaluating federated learning algorithms. They should not be restricted to particular models, unless exploring new applications of federated learning in emerging areas, such as diffusion models or large language models. However, this paper addresses a well-established image classification task and is effective only for the Tsetlin Machine, which limits its practical application.
- The readability of this paper can be further improved. For instance, in line 146, what does the $j$ of $L_j$ stand for, and how to get the definition of the $L_j$ from the definition of $L$?

**Questions:**

- Since FedTMOS uses a non-DNN model, is its scalability being limited by Tsetin Machine? Can it achieve comparable performance when other baseline methods employ stronger networks (e.g., ResNet) on challenging datasets (e.g., Tiny-ImageNet)?

---

> ### Comment · Reviewer_Voa4 · 2024-11-22
> **Reply to Rebuttal**
>
> Thanks for the rebuttal, especially the additional experiments. Since [1] does not provide quantitative results comparing CNN and TM performance, the reviewer still suggest including the performance of 5-layer CNNs and TMs in a centralized setting in Table 1. For applicability, the reviwer acknowledge the potential of TMs as an alternative to DNNs in one-shot FL. However, given that powerful backbones like ViTs can already be employed on some edge devices [2], a deeper discussion on the scalability and performance boundaries of TMs would be beneficial. I would raise my score because of the effort of the additional experiments.
>
> [1] "TMComposites: Plug-and-Play Collaboration Between Specialized Tsetlin Machines", 2023
>
> [2] "FLHetBench: Benchmarking Device and State Heterogeneity in Federated Learning", CVPR 2024

---

### Official Review · Reviewer_Fgwm · 2024-10-27

**Soundness:** 2
**Presentation:** 2
**Contribution:** 3
**Rating:** 6
**Confidence:** 5

**Summary:**

The authors leveraged Testlin Machine to resolve the bottleneck in one-shot federated learning, saving the communication cost and reducing the necessity of using a public dataset. The proposed solution views the one-shot federated learning in a different prospective, in the form of automation machines.

**Strengths:**

1. The idea of introducing Testlin Machine into one-shot federated learning is innovative, aiming to solve the bottleneck of using public datasets.

2. The authors clearly described the background, laying emphasis on Testlin Machine, making the paper self-contained.

3. The authors evaluated the solutions over client numbers of a certain scale, e.g. 20, 50, 80, which is a critical factor in one-shot federated learning.

**Weaknesses:**

1. The reviewer acknowledges the innovation of introducing Testlin Machine, however the motivation for doing so is not well explained. The authors spent certain paragraphs describing  Automation Machine and the mechanisms in machine learning. Nevertheless, how such a mechanism can benefit machine learning and federated learning is not illustrated. Moreover, why the key bottleneck in one-shot federated learning can be resolved is not explained. In other words, the current solution looks like converting a conventional question into a mechanism of a automation machine. For example, it likes a task converting a coding task into Moore Machine in algorithm lectures.

2 Many choices of approaches are not well justified. See more details in the reviewer's questions.

3. The empirical evaluation can be improved. The authors claimed that they used various datasets. However, these are very basic datasets like MNIST,SVHN, and CIFAR10. The reviewer suggested using more complex datasets such as Tiny-ImageNet. For datasets like MNIST, even if we are not doing one-shot federated learning, few epochs and communication rounds are needed to achieve convergence. The effectiveness, particularly in terms of convergence and accuracy, can be correctly justified by using a more complex dataset.

Other minor writing issues:
1. The acronym in the paper is not of common use. OFL is not a common usage for one-shot federated learning. Directly saying one-shot FL is fine. TM is usually referred to Turing Machine.
2. Table 1 and Table 4 are out of bounds.

**Questions:**

1. Why do authors typically introduce Testlin Machine, which is an automation machine rather than leveraging a general reinforcement learning scheme where penalty, reward, and stage changing are involved? What is the motivation for doing so? How different is the solution with general reinforcement learning based one-shot FL? e.g.

2 It is not common to use Gini index to measure data distribution. There are more common solutions. For example, the simplest way is Gaussian Model. But it is possible that clients data are non-i.i.d. In that case, a simple solution is to do some sampling. In some semi-supervised federated learning, uploading hard or soft labels is also fine.  Choosing Gini index is neither a straightforward nor a trivial option. How did you come up with that? And why can authors benefit from that?

---

### Official Review · Reviewer_GkyH · 2024-11-03

**Soundness:** 3
**Presentation:** 2
**Contribution:** 3
**Rating:** 8
**Confidence:** 4

**Summary:**

This paper presents FedTMOS, a compute efficient one-shot Federated Learning (FL) algorithm that leverages Tsetlin Machines. Tsetlin Machines present an alternative to DNNs, known for their low complexity, compute and storage efficiency along with good performance. FedTMOS learns client-specific TMs and derives an aggregated server side TM that enhances class distinction. The aggregation procedure is significantly cheaper than traditional KD based methods while being data-free. The authors show comprehensive empirical results on standard OFL benchmarks under non-IID data.

**Strengths:**

- The application of Tsetlin Machines to OFL is novel and offers an interesting alternative to standard KD based methods which are compute intensive
- The method is data-free
- The authors provide comprehensive evaluations on communication and compute efficiency alongside accuracy which showcase the strength of the approach

**Weaknesses:**

The paper can be improved on several fronts as listed below:
1) The paper offers no discussion on the limitations of Tsetlin Machines and its broader applicability. While TMs are an evolving research area, DNNs are the norm today. Thus, an elaborate discussion of its current limitations will strengthen the paper by well informing the community on its wider applicability. For instance, can TMs be applied to NLP based tasks such as those based on transformer models as of today?
2) A significant portion of the proposed algorithm in Section 4 is explained in sentences, making it difficult to follow without using mathematical references to the quantities being discussed. For instance, equation (4) describes general k-means clustering without reference to actual scaled weights which are being clustered.  Section 4.2.2 uses no mathematical expressions to describe the proposed algorithm. The paper can be greatly improved by defining appropriate notation for quantities being referred to at the beginning of Section 4 and then using this notation throughout while explaining the proposed approach.
3) The paper misses an important baseline, FedFischer [1] which is more compute efficient on the server side as compared to the KD based methods and offers strong accuracy. In general, the paper misses related work involving averaging based schemes such as OT-Fusion [2] and RegMean [3] which offer low server side latency.
4) Lack of theory to justify the performance improvements as compared to the evaluated baselines. Can the authors provide more insights into the accuracy improvements achieved?
5) With the increasing availability of large pre-trained models, conducting OFL starting from a pre-trained initialization is shown to significantly improve performance [1]. How can a TM incorporate pre-trained weights from other TMs trained on large datasets?

[1] Jhunjhunwala, Divyansh, Shiqiang Wang, and Gauri Joshi. "FedFisher: Leveraging Fisher Information for One-Shot Federated Learning." International Conference on Artificial Intelligence and Statistics. PMLR, 2024.

[2] Singh, Sidak Pal, and Martin Jaggi. "Model fusion via optimal transport." Advances in Neural Information Processing Systems 33 (2020): 22045-22055.

[3] Xisen Jin, Xiang Ren, Daniel Preotiuc-Pietro, and Pengxiang Cheng. Dataless knowledge fusion by merging weights of language models. In The Eleventh International Conference on Learning Representations, 2023.

**Questions:**

1) The authors mention using a standard compute node for evaluating server side latency. Does this mean that the node was GPU equipped? It would be unfair to measure the latency of DNN based approaches without using a GPU equipped node.

---

### Meta-Review · Area_Chair_YizU · 2024-12-08

**Metareview:**

This paper introduces FedTMOS, a computationally efficient one-shot Federated Learning (FL) algorithm built on Tsetlin Machines (TMs). Unlike deep neural networks (DNNs), TMs offer low complexity, computational efficiency, and storage savings while maintaining strong performance. The novel application of TMs to one-shot federated learning provides a data-free and compute-efficient alternative to standard knowledge distillation methods. Comprehensive evaluations demonstrate its strengths in accuracy, communication efficiency, and scalability, addressing critical bottlenecks in OFL. The approach is novel, and all the reviewers are positive on the paper, which is why I recommend acceptance of this work.

**Additional Comments On Reviewer Discussion:**

NA

---

### Decision · Program_Chairs · 2025-01-22

Accept (Poster)